# Missing the input: The underrepresentation of plant physiology in global soil carbon research

Sajjad Raza[1], Hannah V. Cooper[1], Nicholas T. Girkin[1], Matthew S. Kent[1], Malcolm J. Bennett[1], Sacha J. Mooney[1], Tino Colombi[1]

[1]School of Biosciences, University of Nottingham, Sutton Bonington, LE12 5RD, United Kingdom

*Correspondence to*: Tino Colombi (tino.colombi@nottingham.ac.uk)

**Abstract.** Plant processes regulating the quantity and quality of soil organic carbon inputs such as photosynthesis, above- and belowground plant growth, and root exudation are integral to our understanding of soil carbon dynamics. However, based on a bibliometric analysis including more than 55 000 scientific papers, we found that plant physiology has been severely
underrepresented in global soil organic carbon research. Less than 10% of peer-reviewed soil organic carbon research published in the last century addressed plant physiological processes relevant to soil carbon inputs. Similarly, plant physiology was overlooked by the overwhelming majority (>90%) of peer-reviewed literature investigating linkages between soil organic carbon, climate change, land use and management. These findings highlight that our understanding of both soil carbon dynamics and the carbon sequestration potential of terrestrial ecosystems is largely built on research that neglects the
fundamental processes underlying organic carbon inputs. We advocate that the active engagement of plant scientists in soil carbon research is imperative to shed light on this blind spot. Long-term interdisciplinary research will be essential to develop a comprehensive perspective on soil carbon dynamics and to inform and design effective policies that support soil carbon sequestration.

## 1 Introduction

Plants and their ability to fix atmospheric carbon dioxide ($CO_2$) through photosynthesis are essential to organic matter buildup in soil (Hirt et al., 2023), the second largest carbon pool on earth (Lal, 2018). The overwhelming majority of organic carbon in soil is derived directly or indirectly from above- and belowground plant residues or rhizodeposits, referring to all organic compounds released by roots (Pausch and Kuzyakov, 2018) (Figure 1). Beyond its critical role in terrestrial carbon sequestration, soil organic carbon supports soil fertility and ecosystem productivity through improved water infiltration and
retention, enhanced soil structure formation, and greater soil biological activity (Lal, 2018). Since the United Nations Conference on Environment and Development (UNCED) held in Rio de Janeiro, Brazil in 1992, soil organic carbon and its fundamental importance in climate change mitigation and adaptation gradually gained importance in the public discussion on sustainable development (Montanarella and Alva, 2015). Recently launched international policy frameworks and initiatives such as the 'European Green Deal' (European Commission, 2019) and the '4 per mille initiative' (Lal et al., 2015) aim to

enhance soil organic carbon levels through adaptations in land use and management. Moreover, the Intergovernmental Panel on Climate Change (IPCC) highlighted the vulnerability of soil organic carbon stocks to environmental disturbances associated with climate change including rising temperatures and extreme weather events such as drought, flooding, or heat waves (IPCC, 2023).

Environmental conditions affect soil organic carbon turnover and stabilisation as well as plants and their physiology, which can lead to feedback with soil carbon turnover through changes in soil moisture, nutrient availability, or soil structure. In addition, plant physiological responses to environmental cues have direct impacts on the quality and quantity of soil carbon inputs. For example, a global meta-analysis encompassing natural and managed ecosystems across various biomes demonstrated that rising temperatures result in a shift of carbon allocation from shoots to roots, particularly in drier climates (Zhou et al., 2022). Similar shifts in carbon allocation from shoots to roots have been reported for wheat in response to decreasing fertilisation intensity (Hirte et al., 2021). Moreover, it has been shown that decreasing fertilisation intensity and water availability increases rhizodeposition rates in arable crops (wheat and maize; Hirte *et al.*, 2018) and temperate tree species (*Picea abies* and *Fagus sylvatica*; Brunn *et al.*, 2022), respectively. Besides affecting carbon allocation between different plant organs and metabolic pathways, environmental conditions such as temperature (Sanaullah et al., 2014; Wang et al., 2012), soil moisture (Sanaullah et al., 2014), nutrient availability, and atmospheric $CO_2$ concentration (Blaschke et al., 2002) alter the biochemical composition of plant tissues and thus the quality of plant litter inputs to soil. Hence, plant physiological responses to climatic conditions or changes in land use and management are absolutely imperative to our understanding of soil carbon dynamics and thus the carbon sequestration potential of terrestrial ecosystems (Figure 1).

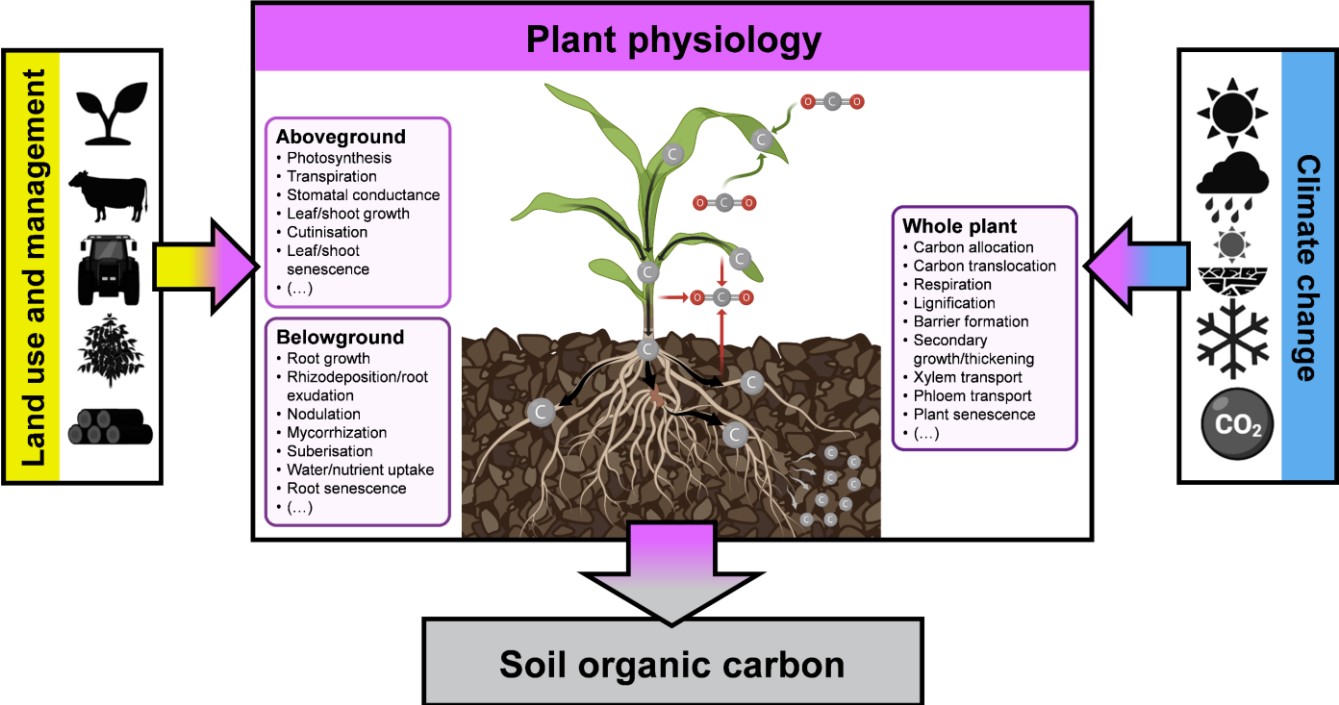

**Figure 1: Conceptual schematic depicting the central role of plants for soil carbon dynamics. Carbon fluxes from the atmosphere into the soil underlying the quantity and quality of soil organic carbon inputs are driven by a suite of plant physiological processes. These physiological processes and their responses to alterations in land use and management or climatic conditions are therefore key to the current and future potential for soil carbon sequestration. Some elements were created with BioRender.com.**

## 2 Soil carbon research largely overlooks plant physiology

Despite the intrinsic linkages between plants, environmental conditions, and soil carbon inputs, plant physiology has been severely underrepresented in global peer-reviewed research on soil organic carbon. We quantified this underrepresentation of plant physiological processes underlying the inputs of organic carbon to soil through bibliometric analyses of data extracted from Web of Science™ (https://www.webofscience.com/), focusing on publication titles, abstracts, keywords, the field, and author keywords (see Appendix for details). To obtain a comprehensive picture on the importance of plant physiology in soil

organic carbon research, we included 64 different plant processes that are directly linked to soil organic carbon inputs (Table S1 in the Supplement).

Our bibliometric analyses revealed that out of 55 207 publications on soil organic carbon published between 1904 and 2024, just 4 855 addressed plant physiology (9%; Figure 2A). To gain additional insight, we grouped the 64 physiological processes into the following three subcategories: aboveground physiology, belowground physiology, and whole plant physiology, i.e.,

processes occurring in above- and belowground plant tissues (Table S2 in the Supplement). This grouping revealed that aboveground physiological processes were represented in 24% of soil organic carbon research addressing plant physiology.

Belowground and whole plant physiological processes were each represented in around 45% of the publications (Figure 2A). More than 95% of the publications on soil organic carbon were published after 1990 and the yearly research output increased exponentially between 1990 and 2024. We observed a parallel temporal trend for peer-reviewed publications on soil organic carbon that addressed plant physiological processes. Therefore, the relative proportion of publications addressing plant physiological processes in the total number of soil organic carbon publications remained approximately constant between 1990 and 2024 (6% to 12%; Figure 2A). Similarly, plant physiology was also largely overlooked in peer-reviewed literature on soil organic carbon modelling (940 out of 13 644 publications, i.e. 7%; Fig. S1 in the Supplement).

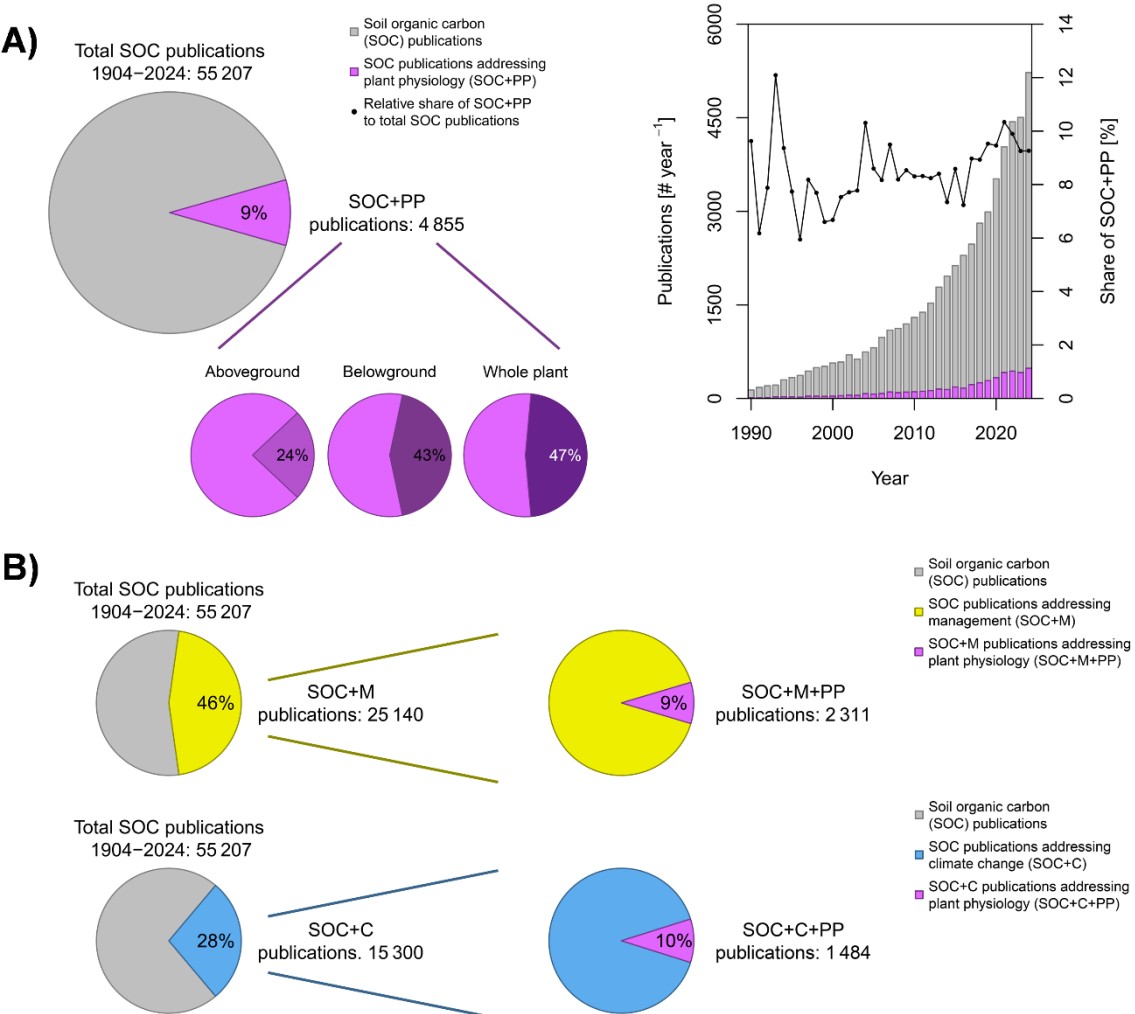

**Figure 2: The representation of plant physiology in global soil organic carbon research. A) Share of soil organic carbon publications addressing plant physiological processes displayed (left) cumulatively from 1904 to 2023 and (right) yearly from 1990 to 2023. Small pie charts in the left panel depict representation of aboveground, belowground, and whole plant physiological processes in soil organic carbon publications addressing plant physiology. B) Soil organic carbon publications that addressed (top) land use and management and (bottom) climate change effects and the share of publications within these two categories addressing plant physiological processes from 1904 to 2023.**

Further bibliometric analyses were conducted to quantify the importance of land use and management and climate change in soil organic carbon research. For these analyses, we included more than 45 keywords to capture relevant land use and management systems and more than 25 keywords to cover climate change and associated environmental conditions (Table S1 in the Supplement). Land use and management was addressed by 46% of the peer-reviewed research on soil organic carbon published between 1904 and 2024 (25 140 out of 55 207 publications), while climate change and concomitant environmental conditions were addressed in 28% of the publications (15 300 out of 55 207 publications; Figure 2B). In contrast to plant physiology, land use and management as well as climate change have been increasingly represented in global research on soil organic carbon. The share of publications on soil organic carbon that addressed land use and management increased from around 35% in the 1990s to almost 50% in the 2020s. Similarly, the share of soil organic carbon publications addressing climate change and associated environmental conditions increased from around 10 to 15% in the early 1990s to more than 30% in the 2020s (Fig. S1 in the Supplement). This increase can likely be attributed to discussions in the public arena on the role of land use and management for climate change mitigation and sustainable development (European Commission, 2019; Lal et al., 2015) that followed the United Nations Conference on Environment and Development (UNCED) in 1992 (Montanarella and Alva, 2015).

However, plant physiology was severely underrepresented in global research elucidating associations between soil organic carbon and land use and management or climate change. Less than 10% of the research covering linkages between soil organic carbon and land use and management (2 311 out of 25 140 publications) and climate change (1 484 out of 15 300 publications), respectively, addressed plant physiological processes (Figure 2B). In the case of climate change, the share of publications addressing plant physiological processes even decreased from around 18% in the 1990s to around 9% in recent years. For land use and management, the percentage of publications addressing plant physiological processes remained around 9% between the 1990s and the 2020s (Fig. S1 in the Supplement). Hence, our bibliometric analyses highlighted that our current understanding of soil carbon dynamics is overwhelmingly based on research that does not account for plant physiological responses to changes in land use and management or climatic conditions. We can only speculate why plant physiology was largely overlooked in soil organic carbon research but the recently reported separation of soil and agricultural sciences in the 1980s (Sigl et al., 2023) may have been a key driver. Ultimately, the staggering underrepresentation of plant physiology in soil organic carbon research reported here (Figure 2; Fig. S1 in the Supplement) severely limits the predictive power of terrestrial carbon models and prevents a comprehensive perspective on the potential for soil carbon sequestration (Fatichi et al., 2019).

## 3. Long-term interdisciplinary research efforts are key

Interdisciplinary research efforts that explicitly integrate plant physiological processes into studies on soil carbon dynamics are urgently needed to develop a more holistic understanding of the drivers underpinning soil carbon sequestration (Hirt et al., 2023). Recent technological and methodological advancements have substantial potential to decipher the complex interactive effects between plant physiological processes and environmental conditions on soil carbon dynamics (Ahkami et al., 2024;

Mueller et al., 2024). For example, the combination of three-dimensional (3D) imaging approaches such as X-ray and positron emission tomography with spectroscopic techniques and carbon isotope tracing facilitate quantifying the flux of photosynthates along the shoot-root-soil axis (Jahnke et al., 2009; Lippold et al., 2023). Moreover, isotope tracing allows to quantify carbon transfer from plants to mycorrhizal fungi and bacteria associated with roots and fungal hyphae (Kaiser et al., 2015; Vidal et al., 2018) and to elucidate the fate of plant litter and rhizodeposits into different soil carbon pools (Cotrufo et al., 2022; Mueller et al., 2024). Thus, combining different cutting-edge techniques offers ample potential to decipher the interactions between plant physiological and soil processes and their impact for soil carbon input. Especially if combined with empirical and mechanistic mathematical models (Ahkami et al., 2024), these approaches enable new mechanistic and predictive insights into the interactions between plant physiological processes and soil carbon dynamics.

However, long-term data obtained at the field and landscape scale is indispensable when quantifying temporal trajectories of soil organic carbon stocks (Smith et al., 2020). We therefore advocate that the regular quantification of plant physiological processes that govern the quality and quantity of soil carbon inputs must become standard in soil surveying, long-term field experiments and observation networks dedicated to soil carbon dynamics. This includes, but not limited to, above- and belowground plant growth (Hirte et al., 2021; Zhou et al., 2022), degrees of lignification and suberisation of plant tissues determining the biochemical quality of plant litter (Blaschke et al., 2002; Wang et al., 2012), and the quantity and composition of rhizodeposits (Brunn et al., 2022; Hirte et al., 2018). Thereby, the assessment of plant physiological processes using drones (Fullana-Pericàs et al., 2022) or satellites (Jonard et al., 2020) may complement and -at least partially- replace laborious ground truth measurements. As for soil organic carbon measurements (Even et al., 2025), standardised protocols to quantify physiological processes underlying soil carbon inputs will be key to facilitate regional and global data synthesis.

Plant scientists including geneticists, physiologists, and ecologists must become an active and integral part of the global soil organic carbon research community. Otherwise, the environmental and genetic drivers underpinning soil carbon inputs will remain a blind spot in our collective understanding of the capacity for soil carbon sequestration (Hirt et al., 2023). Therefore, it is crucial for the global scientific community, including researchers and funding agencies, to recognise the pivotal role of plant physiology in shaping soil carbon dynamics. Without this recognition, our understanding of soil carbon sequestration potential across diverse terrestrial ecosystems will remain incomplete. To accurately model and predict these dynamics— whether through empirical, mechanistic, or geostatistical models—long-term data collection at appropriate spatial and temporal scales is essential (Fatichi et al., 2019; Smith et al., 2020). These models are indispensable for extrapolating interactions across ecosystems and for quantifying effects of climatic conditions and land use and management on plant physiological processes and the resulting impacts on soil carbon dynamics. However, developing such models requires sustained investment in long-term research and improved funding mechanisms that facilitate collaboration among interdisciplinary groups of researchers and relevant stakeholders. Only with these continued efforts can we develop the comprehensive understanding necessary to inform effective policies that support and enhance soil carbon sequestration.

**Appendix: Query design, data extraction and processing**

Comprehensive bibliometric analyses were conducted using the 'advanced search' option in the Web of Science™ data base from Clarivate™ (Web of Science Core Collection; https://www.webofscience.com/). Thereby, the topic search function ("TS") was used to cover publication titles, abstracts, keywords and the field (i.e. keywords plus®), and author keywords. To quantify the share of publications (on soil organic carbon in general or soil organic carbon modelling) addressing plant physiology, land use and management, climate change and associated environmental conditions, or combinations thereof in global research on soil organic carbon, five separate queries were built: i) Soil organic carbon, ii) Modelling, iii) Plant physiological processes, iv) Land use and management, and v) Climate change and associated environmental conditions. For each of the five queries a list of relevant key words and corresponding search terms were defined (i): 2 key words, 11 search terms; ii): 1 key word, 4 search terms; iii): 64 key words, 80 search terms; iv): 47 key words, 66 search terms; v): 28 key words, 41 search terms; Table S1 in the Supplement).

Within each query, the different search terms were connected with the Boolean operator "OR" to ensure the retrieval of all relevant records. Combining the different queries then allowed us to quantify the share of peer-reviewed publications on soil organic carbon addressing plant physiological processes, land use and management, climate change and associated environmental conditions, or combinations thereof. To do so, the search terms of the soil organic carbon query were connected to the search terms of one or several of the other three queries using the Boolean operator "AND". Data searches were conducted on 31 January, 2025 including all records available on Web of Science™ and results were exported as .BIB, .CSV, and .RIS files.

Quality checks of the raw data exported from Web of Science™ were performed with a standardised data filtering pipeline implemented in R version 4.0.2 (R Core Team, 2020) using the package 'bibliometrix' (Aria and Cuccurullo, 2017). First, duplicates and non-English publications were removed. Then, we removed all publications published in 2025 to ensure that only complete years were included. Finally, the data was limited to what is typically considered primary research or scholarly works, i.e. articles, proceedings papers, review articles, early access papers, and data papers (Fig. S2 in the Supplement). These filtering steps reduced the total number of publications included in our analyses by around 5% from 58 004 to 55 207. Data visualisation was performed in R using the packages 'stats' (R Core Team, 2020).

**Data availability**

The bibliometric data is provided in the supplement (Data S1 in the Supplement).

## Author contribution

SR and TC conceived the study. All authors contributed to query design. SR performed the bibliometric analyses with help of MSK. SR and TC wrote the original draft with inputs from HVC. NTG, MSK, MJB, and SJM, contributed to reviewing and editing.

## Competing interests

The authors declare that they have no conflict of interest.

## Acknowledgements

BioRender (https://www.biorender.com/) was used to create elements of Figure 1.

## Financial support

SR, HVC, MJB and SJM received funding from the BBSRC Project Designing Sustainable Wheat (BB/X018806/1). NTG is funded by NERC (NE/X015238/1) and MSK is funded by the Human Frontiers Science Program (RGP0066/2021). TC acknowledges funding from the University of Nottingham (Nottingham Research Fellowship).

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
