# Peer review of "Missing the input: The underrepresentation of plant physiology in global soil carbon research"

_EGUsphere, 2024_

## Referee Comment (RC2)

[referee-annotated manuscript omitted]

---

## Author Response (AR1)

**School of Biosciences**
University of Nottingham
Sutton Bonington Campus
Room C31 Gateway Building
Sutton Bonington
LE12 5RD
United Kingdom

tino.colombi@nottingham.ac.uk

**11 February 2025**

**Revisions of Forum Article by Raza et al.**

Dear Prof. Paul Hallett

Herewith, we submit the revised version of our manuscript "*Missing the input: The underrepresentation of plant physiology in global soil carbon research*" for consideration of publication in *SOIL*.

We would like to thank the two anonymous reviewers for their constructive comments and suggestions that helped to improve the manuscript. Please find our responses to the comments raised by the reviewers below. We have indicated the line numbers where changes to the text have been made, and the corresponding edits are highlighted in yellow in the revised manuscript.

We truly hope that the modifications and revisions made are satisfactory and that the article will be accepted for publication in *SOIL*.

Yours sincerely,

Tino Colombi, PhD (on behalf of all co-authors)

Dear Dr Stefano Manzoni

We thank you for your positive assessment and the constructive comments and suggestions. The responses to your comments and line numbers where we intend to make corresponding changes to the manuscript are listed below. Please note: in the revised version of the manuscript, bibliometric data from 2024 will also be included. Therefore, specific figures on the number of publications will change slightly compared to the original submission, however, overall trends and findings remained the same.

Kind regards

Tino Colombi, on behalf of all co-authors

Raza and co-authors discuss the lack of plant physiological processes in soil carbon cycling research. This issue is highly relevant as much of soil carbon cycling is driven by plant activity, in particular in the rhizosphere or related to plant symbionts. Their results are clear—the large majority of soil carbon studies do not include plant physiology even though a plant physiological perspective could provide novel or more mechanistic insights on soil carbon processes.

***Response: Thank you for this positive and encouraging assessment.***

My main concern is that only a few examples of plant physiological processes that matter for soil carbon cycling are mentioned, and similarly not many examples of empirical approaches to characterize plant physiology are reviewed and discussed. For example, what physiological processes are relevant for plant-mycorrhizae interactions, which in turn affect soil carbon processes? What is the fate of rhizodeposits in soil fractions (Cotrufo et al., 2022)? How can isotopic tracer methods be used (only briefly mentioned now)? To make this work more complete, I would suggest going a bit deeper into key processes (or groups of processes) and also check with the bibliometric analysis is some of them are missing more than others. Examples of process groups could be: rhizodeposits, carbon allocation to roots or mycorrhizae, litter quality (chemistry, element ratios), root properties affecting soil structure (aggregation, microporosity).

***Response: In our opinion, this comment entails three distinct points. Please find our detailed responses below.***

***1) Including more plant physiological processes: As indicated in the title of our manuscript (and at multiple points throughout the text), we want to focus here on the role of plants for soil carbon inputs. Despite being by far the most important group of terrestrial primary producers, plants and their physiology are severely underrepresented in the ever-growing body of soil carbon literature (Figure 2). Moreover, we found that there is no trend towards alleviating this underrepresentation. In certain cases, this underrepresentation even increased in recent years (see Supplemental Figure S1 of the original submission). We think that our article requires a clear (and somewhat narrow) focus on plant physiological processes linked to soil carbon inputs to make it evident that our current understanding of soil carbon dynamics is overwhelmingly built on work that overlooks the fundamental processes underlying soil carbon inputs. Therefore, we think that our bibliometric analyses must target plant physiological processes linked to soil carbon inputs. Yet, we do not deny in any way that plant physiology has implications for other parts of the soil carbon cycle,***

*e.g., on soil carbon turnover through changes in soil moisture, nutrient availability, or structure. To make this clear, the first sentences of the second paragraph in the introduction will be modified to: "Environmental conditions affect soil organic carbon turnover and stabilisation as well as plants and their physiology, which can lead to feedback with soil carbon turnover through changes in soil moisture, nutrient availability, or soil structure. In addition, plant physiological responses to environmental cues have direct impacts on the quality and quantity of soil carbon inputs" (L34-38).*

*Taking these points into consideration, we agree that our original list of plant physiological processes should be extended to ensure broader coverage of relevant processes. We will therefore add 40 additional key words to our bibliometric query (please note: "carbon translocation" was included in the query used for the original submission, see: Supplemental Table S1 of the original submission). These key words will cover/add to three distinct categories, i.e. i) interactions of plants with belowground biota, ii) biochemical composition of soil carbon inputs, and iii) water and nutrient uptake and translocation. The following key words will be added in the revised version of the manuscript:*

- *plant carbon transfer,*
- *rhizodeposition,*
- *nodulation, rhizobia colonisation,*
- *mycorrhization, fungal colonisation, mycorrhizal colonisation*
- *cutinisation,*
- *tissue stoichiometry, leaf stoichiometry, root stoichiometry, shoot stoichiometry, plant stoichiometry, litter stoichiometry, rhizodeposits stoichiometry, root exudate stoichiometry,*
- *tissue element ratio, leaf element ratio, root element ratio, shoot element ratio, plant element ratio, litter element ratio, rhizodeposits element ratio, root exudate element ratio,*
- *tissue composition, leaf composition, root composition, shoot composition, plant composition, litter composition, rhizodeposits composition, root exudate composition,*
- *plant water uptake, plant water acquisition, plant water translocation,*
- *xylem transport, phloem transport,*
- *plant nutrient uptake, plant nutrient acquisition, plant nutrient translocation, plant nutrient absorption,*

*Please note: If British and American spelling differ (e.g. colonisation/colonization), both versions will be included in the bibliometric search. This updated query will contain 64 keywords (as compared to 24 in the original submission) and 80 corresponding search terms. The inclusion of these additional search terms changed the results only marginally: Out of 55 207 peer-reviewed publications on soil organic carbon published between 1904 and 2024, plant physiology was address by only 4 855 publications, corresponding to 8.8%. In the revised version of the manuscript, these additional key words will be included and listed in Supplemental Table S1. Furthermore, we will adapt the main text (L60-63, L97-100), the appendix (L154) and the figures accordingly (Figure 2, Supplemental Figures S1 and S2).*

*2) Grouping of plant physiological processes to gain additional, more nuanced insights: Thank you for this valuable suggestion. We agree that grouping the processes will provide additional insights, thereby highlighting key knowledge gaps and future research needs. We*

*suggest dividing the 64 keywords into the following three groups: i) aboveground/shoot physiology, ii) belowground/root physiology, and iii) whole plant physiology (keywords and corresponding search terms of the three groups will be summarised in Supplemental Table S2 of the revised version of the manuscript). Performing this analysis revealed that 24% of the peer-reviewed papers on soil organic carbon that address plant physiology included aboveground physiological processes, 43% included belowground processes, and 74% included whole plant processes. Thus, the underrepresentation of plant physiology in soil carbon research is particularly severe for aboveground processes. In the revised version of the manuscript, these figures will be included into Figure 2 (L74). The grouping of the different processes into the above-mentioned categories and the corresponding results will be presented in the main text as "To gain additional insight, we grouped the 64 physiological processes into the following three subcategories: aboveground physiology, belowground physiology, and whole plant physiology (Supplemental Table S2). This grouping revealed that aboveground physiological processes were represented in 24% of soil organic carbon research addressing plant physiology. Belowground and whole plant physiological processes were each represented in around 45% of the publications (Figure 2A)." (L63-67).*

*3) More information on empirical approaches to quantify plant physiological processes underlying soil carbon inputs: Reviewer 2 made a similar comment and suggested to elaborate a bit more on methods to quantify plant physiological processes relevant to soil carbon inputs. To address these comments of both reviewers, we will provide additional information on empirical approaches that are suitable to gain a better understanding of the linkages between plant physiology and soil carbon inputs, both under controlled conditions and at the field and landscape scale. In detail, "Moreover, isotope tracing allows to quantify carbon transfer from plants to mycorrhizal fungi and bacteria associated with roots and fungal hyphae (Kaiser et al., 2015; Vidal et al., 2018) and to elucidate the fate of plant litter and rhizodeposits into different soil carbon pools (Cotrufo et al., 2022; Mueller et al., 2024). Thus, combining different cutting-edge techniques offers ample potential to decipher the interactions between plant physiological and soil processes and their impact for soil carbon input." (L115-119) and "Thereby, the assessment of plant physiological processes using drones (Fullana-Pericàs et al., 2022) or satellites (Jonard et al., 2020) may complement and -at least partially- replace laborious ground truth measurements." (L128-130) will be added to the revised version of the manuscript.*

Most of the examples of plant physiological processes in the manuscript deal with direct plant effects on soil carbon cycling, but there are also indirect effects that might be worth mentioning. For example, plants affect soil carbon cycling indirectly via changes in the soil microenvironment (especially soil moisture), and such effects are mediated by plant physiology (in the example of soil moisture, via stomatal regulation, water redistribution through roots etc.).

*Response: As we outline in greater detail in our response above, we do acknowledge that effects of plants on soil carbon dynamics are not exclusively direct but indirect effects as those mentioned here also play a significant role (as we will state in L34-38 of the revised version of the manuscript). However, the focus of this manuscript in on soil carbon inputs and the underlying physiological processes, which can for the large part be classified as rather direct effect of plants on soil carbon dynamics.*

Moreover, I wonder if the gap between soil-focused and plant-focused studies extends also to theoretical or modelling papers. For example, there is a rich literature on linkages between roots and soils via rhizodeposition or symbiotic interactions (Cheng, 1999; Darrah, 1991; Franklin et al., 2014; Smith and Wan, 2019; Zelenev et al., 2000), but I would expect the large majority of soil carbon cycling models focus on microbial or soil faunal processes and only consider plants a source of carbon and nutrients (without accounting for plant physiology).

*Response: We have added an additional query to quantify the representation of plant physiology in peer-reviewed literature addressing soil carbon modelling. The analysis yielded very similar results as observed when considering all soil carbon research (plant physiological processes were addressed in ca. 7% of papers). We will add the following sentence to the revised version of the manuscript to present these results: "Similarly, plant physiology was also largely overlooked in peer-reviewed literature on soil organic carbon modelling (940 out of 13 644 publications, i.e. 7%; Supplemental Figure S1)" (L71-73). Furthermore, we will summarise these results in a Supplemental Figure (Supplemental Figure S1), describe the query in the Appendix (L151, L154), and summarise the key words and corresponding search terms in Supplemental Table S1.*

Specific comments

Figure 1: rather generic representation of soil-plant interactions, nearly without plant physiology. I would include more graphical representations of the physiological processes considered relevant (and overlooked)—e.g., there is no mention of nutrient uptake and re-cycling via litter with feedbacks on carbon cycling; mycorrhizae and microbes in the rhizosphere are linked to plants in different ways. Indirect effects of plant physiology on soil carbon cycling are also not shown.

*Response: We will adapt the figure and add a list of key processes underpinning soil carbon inputs following the same grouping into aboveground, belowground, and whole plant physiological processes as described above. As we outline above, we would like to keep a clear focus on the role of plant physiological processes for soil carbon inputs.*

L59: what does "address" mean? Probably hard to say based on bibliometric analysis, but by reading the papers you found, did you notice if plant physiology is perceived as a driver of soil processes, or rather a consequence of soil processes? These different perspectives could add some nuance to the results presented here.

*Response: As suggested in this comment, it is not possible with bibliometric analyses to evaluate whether the papers look at plant physiological processes as drivers of soil processes or consequences of soil processes. As we describe in the Appendix, bibliometric analyses as done here are based on publication titles, abstracts, keywords and the field, and author keywords. Thus, "address" is in our opinion the most suitable and correct term to use here. To clarify this, we will adjust one sentence in the main text of the revised manuscript to: "We quantified this underrepresentation through bibliometric analyses of data extracted from Web of Science™ (https://www.webofscience.com/) focusing on publication titles, abstracts, keywords, the field, and author keywords (see Appendix for details)." (L56-59).*

*Nevertheless, while we agree that it would be interesting to quantify the percentages of papers treating plant physiological processes as a cause and an effect, respectively, we do not think that this will significantly strengthen the main finding (i.e. the fact that plant physiology and thus the fundamental processes underlying soil carbon inputs are overlooked in soil carbon research). Moreover, we think that grouping the physiological processes into aboveground, belowground and whole plant processes as described in our response to the first comment adds more nuance on the role of plant physiology in soil carbon research.*

L96: I don't disagree with the point raised here, but I wonder if soil scientists perhaps focus on the net result of plant physiological processes on soils, rather than those processes per se. For example, in virtually all soil carbon balance calculations litterfall is an input, but it might be less important what specific physiological process led to that amount of litter—e.g., higher litterfall could be due to higher photosynthesis or lower respiration.

*Response: We also think that the focus in soil science (and related fields) is on the net result of plant physiological processes and not on the processes per-se. This, however, is a very static depiction of plants. To develop a more mechanistic and holistic understanding of soil carbon dynamics, this must change, the highly dynamic and responsive nature of plants must be acknowledged. As outlined by us here and others elsewhere (e.g. Fatichi et al., 2019), plants and their physiological responses to environmental cues are key to terrestrial carbon cycling and thus our ability to predict current and future soil carbon sequestration potential.*

References

Cheng, W. X.: Rhizosphere feedbacks in elevated CO2, Tree Physiol., 19, 313–320, 1999.

Cotrufo, M., Haddix, M., Kroeger, M., and Stewart, C.: The role of plant input physical-chemical properties, and microbial and soil chemical diversity on the formation of particulate and mineral-associated organic matter, Soil Biol. Biochem., 168, https://doi.org/10.1016/j.soilbio.2022.108648, 2022.

Darrah, P. R.: Models of the Rhizosphere .1. Microbial-Population Dynamics around a Root Releasing Soluble and Insoluble Carbon, Plant Soil, 133, 187–199, 1991.

Franklin, O., Naesholm, T., Hoegberg, P., and Hoegberg, M. N.: Forests trapped in nitrogen limitation - an ecological market perspective on ectomycorrhizal symbiosis, New Phytol., 203, 657–666, https://doi.org/10.1111/nph.12840, 2014.

Smith, G. R. and Wan, J.: Resource-ratio theory predicts mycorrhizal control of litter decomposition, New Phytol., 223, 1595–1606, https://doi.org/10.1111/nph.15884, 2019.

Zelenev, V. V., van Bruggen, A. H. C., and Semenov, A. M.: "BACWAVE," a spatial-temporal model for traveling waves of bacterial populations in response to a moving carbon source in soil, Microb. Ecol., 40, 260–272, 2000.

Dear Dr. Grant A. Campbell

We thank you for your positive assessment and the constructive comments and suggestions. The responses to your comments and line numbers where we intend to make corresponding changes to the manuscript are listed below. Please note: in the revised version of the manuscript, bibliometric data from 2024 will also be included. Therefore, specific figures on the number of publications will change slightly compared to the original submission, however, overall trends and findings remained the same.

Kind regards

Tino Colombi, on behalf of all co-authors

**General Comments**

Really interesting piece of research highlighting the lack of awareness into plant physiology in global soil carbon research. The authors have shown strong understanding in where plant physiology could be utilised more in the literature and have presented in clear fashion the process they undertook in evaluating and assessing this as part of a review topic. I also thought the graphics were really well presented on the whole and were easy to interpret (minus one small adjustment).

At this moment, only minor revisions are required and some of these are seen as technical/discussion ideas for the authors to consider. I attach my comments of the manuscript under supplementary PDF.

I place my comments under the particular headings also in order to help the authors shape the revised version of which I am happy to look over again if required. I am also happy to take any comments from the authors should they be not clear of my thoughts.

***Response: Thank you for your positive assessment of our manuscript.***

**Specific comments and technical corrections**

Line 1 - Suggest title change to "Missing the input: The lack of awareness in plant physiology in global soil carbon research

***Response: Thank you for this suggestion. However, we would like to keep the original title since "underrepresentation" better reflects the findings we obtained from the bibliometric analyses. I.e. bibliometric analyses allow to quantify the representation of (in our case) plant physiology in soil carbon research whereas the reasons underlying this representation or underrepresentation remain speculative (as we also state in the manuscript, L92-94). Ultimately, we do not know if the underrepresentation we report here is caused by a lack of awareness or other reasons like methodological challenges or a lack of communication and exchange between the soil and plant science communities.***

Line 13 - land use and management. Remove the first "and".

***Response: Will be adapted as suggested (L13)***

Line 13/14 - I would reword this a little bit. Something like "understanding of both soil carbon dynamics and subsequently carbon sequestration potential"

*Response: This will be adapted following this suggestion to "These findings highlight that our understanding of both soil carbon dynamics and the carbon sequestration potential of terrestrial ecosystems is largely built on research that neglects the fundamental processes underlying organic carbon inputs" (L13-15).*

Line 15 - "continued? dynamic? ever developing? Suggest using one of these words to improve this sentence

*Response: We do not think that adding an additional word/adjective to this sentence would strengthen the message. In our opinion, "...the active engagement..." by itself conveys a clear and strong message. Thus, we would like to keep the original phrasing.*

Line 17 - and design? (when talking about implementing policies)

*Response: Will be adapted as suggested to "Long-term interdisciplinary research will be essential to develop a comprehensive perspective on soil carbon dynamics and to inform and design effective policies that support soil carbon sequestration." (L16-18)*

Line 22 - Bolden Figure 1 text

*Response: We followed the formatting style of SOIL, i.e., in-text references to figures and tables in* "Standard Times New Roman"*. We will stick to this formatting style in the revised version of the manuscript.*

Line 31 - such as? Flooding? Drought? Other?

*Response: We will add "... such as drought, flooding, or heat waves" in the revised version of the manuscript (L32).*

Line 32 - I would reword this sentence a bit more clearly. Something like "Environmental conditions not only influence the stabilization of soil organic carbon but also significantly affect plant physiology, which in turn impacts the quality and quantity of carbon inputs to the soil."

*Response: We will adapt this sentence in response to this comment and a comment raised by Reviewer 1 to "Environmental conditions affect soil organic carbon turnover and stabilisation as well as plants and their physiology, which can lead to feedback with soil carbon turnover through changes in soil moisture, nutrient availability, or soil structure. In addition, plant physiological responses to environmental cues have direct impacts on the quality and quantity of soil carbon inputs." in the revised version of the manuscript (L34-38).*

Line 32 - 33 - This sentence could do with a rejig as at present it looks clunky. Suggest something like ""For instance, a global meta-analysis encompassing both natural and managed ecosystems across various biomes demonstrated that rising temperatures result in a shift of carbon allocation from shoots to roots, particularly in drier climates"

*Response: Following this suggestion, we will adapt this sentence to "For example, a global meta-analysis encompassing natural and managed ecosystems across various biomes demonstrated that rising temperatures result in a shift of carbon allocation from shoots to roots, particularly in drier climates" in the revised version of the manuscript (L37-38).*

Line 40 - you mention temperature but what about precipitation or snowmelt? Suggest adding a source or two to discuss other conditions

*Response: Thank you for this suggestion. We will add soil moisture as another example of environmental conditions that can alter the biochemical composition of plant tissue (L44). The following reference will be added to support this statement: Sanaullah, M. et al. Effects of drought and elevated temperature on biochemical composition of forage plants and their impact on carbon storage in grassland soil, Plant Soil, 374, 767–778".*

Line 43 - Bolden Figure 1

*Response: We followed the formatting style of SOIL, i.e., in-text references to figures and tables in* "Standard Times New Roman"*. We will stick to this formatting style in the revised version of the manuscript.*

Line 45 - 51 - I would just keep Figure 1 in this text as bold only. I am also worried that we have too much text here to discuss the figure., Suggest, if possible to simplify the text.

*Response: We followed the formatting style of SOIL, i.e., figures and tables captions in* **"Bold Times New Roman"***. We will stick to this formatting style in the revised version of the manuscript. We will shorten the caption to "Figure 1: Conceptual schematic depicting the central role of plants for soil carbon dynamics. Carbon fluxes from the atmosphere into the soil underlying the quantity and quality of soil organic carbon inputs are driven by a suite of plant physiological processes. These physiological processes and their responses to alterations in land use and management or climatic conditions are therefore key to the current and future potential for soil carbon sequestration. Some elements were created with BioRender.com." (L50-54).*

Line 54 - Underrepresented as not being discussed in science overall too or just with respect to purely soil organic carbon? Perhaps make this distinction clearer.

*Response: We refer here to the underrepresentation of plant physiological processes in soil carbon research. In our opinion, our formulation "…plant physiological processes have been severely underrepresented in global peer-reviewed research on soil organic carbon." clearly*

*indicates this since we explicitly mention "global peer-reviewed research on soil organic carbon". Therefore, we would like to stick to the original formulation.*

Line 57-58 - Bolden Supplementary Table S1

*Response: We followed the formatting style of SOIL, i.e., in-text references to figures and tables in* "Standard Times New Roman"*. We will stick to this formatting style in the revised version of the manuscript.*

Line 59 - I would place the 3,907 out of 49.,971 here with (8%) in brackets to make this a bit clearer and to make sure the reader is drawn purely to just Figure 2A. "revealed that out of 49,971 publications, just 3,907 (8%)..." Also bolden Figure 2A

*Response: We will adapt the text according to this suggestion to "Our bibliometric analyses revealed that out of 55 207 publications on soil organic carbon published between 1904 and 2023, just 4 855 (i.e. 9%) addressed plant physiology (Figure 2A)" in the revised version of the manuscript (L62-63).*

*Moreover, as above, we followed the formatting style of SOIL, i.e., in-text references to figures and tables in* "Standard Times New Roman"*. We will stick to this formatting style in the revised version of the manuscript.*

Line 64- Bolden Figure 2A

*Response: We followed the formatting style of SOIL, i.e., in-text references to figures and tables in* "Standard Times New Roman"*. We will stick to this formatting style in the revised version of the manuscript.*

Line 65 - 69 Perhaps suggest a different colour here, in Graphic A, to accommodate for colour blind people/photocopy/printing? Avoid greens if possible. I would keep only Figure 2 boldened and keep rest of text as normal text

*Response: Thank you for this remark. We will change the colour scheme of Figure 2 (as well as Figure 1 and the Supplemental Figures S1 and S2) replacing green with light purple. Moreover, as above, we followed the formatting style of SOIL, i.e., figures and tables captions in* "Bold Times New Roman" *and we will stick to this formatting style in the revised version of the manuscript.*

Line 73 - Bolden Supplementary Table S1

*Response: We followed the formatting style of SOIL, i.e., in-text references to figures and tables in* "Standard Times New Roman"*. We will stick to this formatting style in the revised version of the manuscript.*

Line 75 - Bolden Figure 2B

*Response: We followed the formatting style of SOIL, i.e., in-text references to figures and tables in* "Standard Times New Roman"*. We will stick to this formatting style in a revised version of the manuscript.*

Line 80 - Bolden Supplementary Figure S1

*Response: We followed the formatting style of SOIL, i.e., in-text references to figures and tables in* "Standard Times New Roman"*. We will stick to this formatting style in a revised version of the manuscript.*

Line 84 - A couple of policy items should be noted here to potentially add onto this: The European Green Deal; EU Soil Thematic Strategy; UN Sustainable Development Goals; Paris Agreement; EU CAP reforms perhaps as well?

*Response: We agree that these policy items are relevant for soil carbon sequestration and related topics. However, these are all relatively recent developments (last 10 years or less) occurring during (and not before) the increase in the representation of climate change and land use and management in global soil organic carbon research described in L86-91 and shown in Supplemental Figure S2. Thus, in contrast to UNCED held in 1992, these policies cannot explain these trends, but they are possibly a consequence of these trends. Therefore, we do not think that the policy programs mentioned in this comment should be listed here and we would like to keep the original phrasing in the revised version of the manuscript.*

Line 87 - Bolden Figure 2B

*Response: We followed the formatting style of SOIL, i.e., in-text references to figures and tables in* "Standard Times New Roman"*. We will stick to this formatting style in a revised version of the manuscript.*

Line 90 - Bolden Supplementary Figure S1 - change from Supplemental

*Response: We followed the formatting style of SOIL i.e., in-text references to figures and tables in* "Standard Times New Roman"*. We will stick to this formatting style in a revised version of the manuscript. Moreover, we changed all in-text references to supplemental display items to "Table SX/Fig. SX in the Supplement" following the standard of SOIL*

Line 93 - this is definitely true in Europe. Perhaps add more sources to back up this point?

*Response: We are unfortunately not aware of other academic papers that systematically looked at this divide of plant and soil sciences. Hence, we cannot add additional references. Moreover, we explicitly mention that we "... can only speculate ..." (L92). With this formulation, we clearly indicate that this might not be the actual or only reason underling*

*the underrepresentation of plant physiology in soil carbon research. Thus, backing up this speculative claim with more references (if there are any) may not be appropriate.*

Line 95 - Bolden Figure 2 and Supplementary Figure S1

*Response: We followed the formatting style of SOIL i.e., in-text references to figures and tables in* "Standard Times New Roman"*. We will stick to this formatting style in a revised version of the manuscript.*

Line 102 - Add (3D) next to three-dimensional

*Response: We will adapt the text as suggested in a revised version of the manuscript (L113).*

Line 103- Suggest the use of FAPAR as well as a possible option for a toolset - https://en.wikipedia.org/wiki/Fraction_of_absorbed_photosynthetically_active_radiation

*Response: This comment is linked to point raised by Reviewer 1 asking us to provide more information on empirical approaches to assess plant physiological processes that are relevant for soil carbon inputs. In our opinion, the approach suggested here (FAPAR) is part of a wider set of approaches that rely on spectral data obtained from air- or space-borne sensors. To provide a general perspective on the use of spectral sensors, we suggest adding the following sentence to the revised version of the manuscript: "Thereby, the assessment of plant physiological processes using drones (Fullana-Pericàs et al., 2022) or satellites (Jonard et al., 2020) may complement and -at least partially- replace laborious ground truth measurements." (L128-130).*

Line 104 - Break up a bit what we mean by mathematical models? I take it we mean classification and regression approaches? Machine learning overall?

*Response: As we mention in the last paragraph of the main text, we refer to a wide range of modelling approaches (empirical/statistical, as well as mechanistic models; Please see L118-120 in the original submission). To clarify this, we will adapt the text to "Especially if combined with empirical and mechanistic mathematical models..." in the revised version of the manuscript (L119-120).*

Line 109-110 - Agree. Suggest noting the use of MRV standardisation as a potential concept for this (Monitoring, Reporting and Verification)

*Response: We are not sure if we understand this suggestion correctly. We fully agree that standardisation in the acquisition of plant physiological data will be key to facilitate data synthesis. To highlight this, we will add "As for soil organic carbon measurements (Even et al., 2025), standardised protocols to quantify physiological processes underlying soil carbon inputs will be key to facilitate regional and global data synthesis." to the end of this paragraph (L130-131). We hope this change is satisfactory.*

Line 124 - Investment in scientific funding but also in terms of industry interest too I would argue. Especially for working farming environment

*Response: We agree with this and even think that this includes not only farmers but all relevant stakeholder groups. Therefore, we will adapt this sentence to "However, developing such models requires sustained investment in long-term research and improved funding mechanisms that facilitate collaboration among interdisciplinary groups of researchers and relevant stakeholders" (L143) in the revised version of the manuscript.*

**Additional comments (do not to be addressed but should be considered)**

Line 41 - Has there been any discussion about the effect of seasonality in this work? Might be good to address or look at contrasts between even summer and winter months.

*Response: We agree that seasonality can matter greatly, especially given the tight coupling between season (and concomitant weather conditions), phenology, and different plant physiological processes underpinning soil carbon inputs. However, we believe that these interactions are a bit out of the scope of our manuscript and therefore prefer to keep the text as concise as possible in order to convey a clear message.*

Line 62 - Does this suggest an increased awareness in plant physiological interactions with soil organic carbon OR also greater accessibility to access more information/data? Something to think about perhaps rather than to address directly.

*Response: Thank you for this.*

Line 118 – 120 - Should this become a requisite in future soil surveying? European activities such as LUCAS should in my view look at this but also the development and movement of the EU Soil Monitoring Law could play a factor here in whether this could be effectively achieved. I would also argue that the need for continued development and expert knowledge should be being passed over across interested people.

*Response: Thank you for this suggestion. We fully agree and will therefore explicitly mention soil surveying alongside long-term field experiments and observation networks in the revised version of the manuscript, i.e., "We therefore advocate that the regular quantification of plant physiological processes that govern the quality and quantity of soil carbon inputs must become standard in soil surveying, long-term field experiments and observation networks dedicated to soil carbon dynamics..." (L124).*